# Acquisition of chromosome instability is a mechanism to evade oncogene addiction

Lorena Salgueiro[1], Christopher Buccitelli[2,†], Konstantina Rowald[1], Kalman Somogyi[1], Sridhar Kandala[1], Jan O Korbel[2] & Rocio Sotillo[1,3,*]

## Abstract

Chromosome instability (CIN) has been associated with therapeutic resistance in many cancers. However, whether tumours become genomically unstable as an evolutionary mechanism to overcome the bottleneck exerted by therapy is not clear. Using a CIN model of Kras-driven breast cancer, we demonstrate that aneuploid tumours acquire genetic modifications that facilitate the development of resistance to targeted therapy faster than euploid tumours. We further show that the few initially chromosomally stable cancers that manage to persist during treatment do so concomitantly with the acquisition of CIN. Whole-genome sequencing analysis revealed that the most predominant genetic alteration in resistant tumours, originated from either euploid or aneuploid primary tumours, was an amplification on chromosome 6 containing the cMet oncogene. We further show that these tumours are dependent on cMet since its pharmacological inhibition leads to reduced growth and increased cell death. Our results highlight that irrespective of the initial CIN levels, cancer genomes are dynamic and the acquisition of a certain level of CIN, either induced or spontaneous, is a mechanism to circumvent oncogene addiction.

**Keywords** breast cancer; Chromosome instability; cMet; mouse models; resistance

**Subject Categories** Cancer; Chromatin, Transcription & Genomics

See also: **D Bronder & SF Bakhoum** (March 2020)

## Introduction

Chromosome instability (CIN) is a widespread phenomenon in malignancies characterized by the inability of a cell to maintain its diploid chromosome number, leading to a state of aneuploidy. Advances in DNA sequencing technologies have allowed for the high-throughput visualization of tumour genomes and revealed the pervasiveness and diversity of aneuploidy across cancers (Cancer Genome Atlas Research Network *et al*, 2013; Zack *et al*, 2013).

Indeed, 90% of solid and 75% of hematopoietic tumours display aneuploidy (Duijf *et al*, 2013) and are thus thought to have gone through periods of CIN to obtain it. Though CIN is often reported to cause whole or partial chromosome changes, it is also known to cause smaller focal somatic copy number alterations (SCNAs) including amplifications and deletions (Janssen *et al*, 2011; Burrell *et al*, 2013).

Although many SCNAs are clearly pro-tumorigenic (e.g. lower expression of TP53 via whole chromosome loss or focal amplification of KRAS), the majority are thought to be detrimental to the cell (Sansregret & Swanton, 2017). High levels of CIN and thus high rates of SCNA generation may lead to imbalanced expression of proteins encoded on the affected DNA regions, endangering the survival of a tumour's lineage. It is becoming clear that an optimal level of CIN can be achieved that is tolerated by tumour cells while promoting diversification of subclones and facilitating adaptation to selective pressures (e.g. drug treatment) during tumour development (Rowald *et al*, 2016).

Despite recent advances in detection and therapy, breast cancer remains the second leading cause of cancer-related death in women and approximately 20% of breast cancer patients develop recurrent disease following treatment. Therefore, understanding the mechanism by which breast tumours recur and develop therapeutic resistance is critical. Several mechanisms have been observed to date such as activating point mutations in PIK3CA, loss of PTEN or overexpression of cMET (Nagata *et al*, 2004; Berns *et al*, 2007; Shattuck *et al*, 2008).

Mouse models of human cancer provide a suitable setting to look at the molecular and temporal dynamics of tumour recurrence via evolution of oncogene independent subclones. In a recent study, using a mouse model of Braf-induced melanoma, Kwong *et al* (2017) show that under strong selective pressure, genetically stable tumours acquire treatment resistance by mutating and thereby reactivating the initiating oncogenic pathway whereas genomically unstable tumours acquire broad whole chromosome aneuploidies which presumably afford their oncogene independence via a yet unidentified mechanism. Whether this is a general phenomenon for all cancer types or whether it only applies to this model is not known.

1 Division of Molecular Thoracic Oncology, German Cancer Research Center (DKFZ), Heidelberg, Germany
2 Genome Biology Unit, European Molecular Biology Laboratory (EMBL), Heidelberg, Germany
3 Translational Lung Research Center Heidelberg (TRLC), German Center for Lung Research (DZL), Heidelberg, Germany
*Corresponding author. Tel: +49 6221 42 3691; E-mail: r.sotillo@dkfz-heidelberg.de
†Present address: Max Delbrück Center for Molecular Medicine, Berlin, Germany

Mad2 is a central component of the spindle assembly checkpoint responsible for ensuring proper separation of sister chromatids. Its overexpression is commonly found in human cancers and leads to the hyperstabilization of kinetochore-microtubule attachments that can result in mitotic arrest and improper correction of erroneous attachments causing lagging chromosomes, misalignments and consequently, aneuploidy (Rowald *et al*, 2016). Moreover, this ongoing CIN induced by Mad2 overexpression can circumvent oncogene addiction, compromising the effectiveness of targeted therapies thereby facilitating tumour relapse and persistence in lung and breast cancer models (Sotillo *et al*, 2010; Rowald *et al*, 2016). Using a doxycycline-inducible mouse model of mutant Kras, we showed that mice develop multiple invasive mammary adenocarcinomas within 147 days, while the additional overexpression of Mad2 delays tumour onset (221 days) and increases the levels of CIN in the resultant breast tumours (Rowald *et al*, 2016). Moreover, downregulation of Kras or Kras and Mad2 in these fully formed breast tumours leads in some cases to the development of resistance (Rowald *et al*, 2016).

These results suggest that withdrawal of the oncogenic driver event in the presence of ongoing CIN would not be deleterious as this would enable further gain or loss of chromosomal regions that could sustain tumour growth. Here, we show that both low CIN and high CIN breast cancer models may resist oncogene withdrawal by activating effectors in downstream or parallel pathways to the initiating oncogene and further characterize the temporal effects of CIN on disease progression. Furthermore, we demonstrate that the few initially low CIN tumours that manage to resist oncogene withdrawal do so by acquiring CIN and achieving similar SCNA levels as initially high CIN tumours when faced with the pressure of oncogene withdrawal.

# Results

## Oncogene independent Kras and Kras/Mad2 tumours show high levels of chromosomal instability

Doxycycline-inducible mouse models of Kras$^{G12D}$ (K) and Kras$^{G12D}$/ Mad2 (KM) allow us to mimic targeted therapy since doxycycline withdrawal leads to complete silencing of the transgenes. Downregulation of Kras expression or Kras and Mad2 in fully formed mammary tumours results in their regression to a nonpalpable state in the vast majority of the cases. However, 6.6% of K and 21.3% of KM tumours did not fully regressed and were able to continue growing (Rowald *et al*, 2016). This indicates that, despite the initial detrimental effect of Mad2 overexpression, these resulting highly unstable tumours have increased chances to develop therapy resistance. Moreover, supporting the idea that chromosomal instability at primary tumour level confers advantages in strong pressure environments such as during targeted therapy, KM tumours needed on average 46 days to grow back while K tumours needed 93 days.

To evaluate chromosome segregation fidelity in these tumours that did not regress upon doxycycline withdrawal or that partially regressed and resumed growth (termed non-regressed), we performed time-lapse microscopy of the non-regressed mammary tumour-derived cells. Interestingly, we found that, while KM primary tumours were significantly more unstable than K primary

tumours (Rowald *et al*, 2016), K and KM non-regressed tumours showed similar percentages of mitotic errors, surpassing in both cases the already high level of CIN found in KM primary tumours (Fig 1A and B). Regardless of the genotype, we observed lagging chromosomes, chromosome bridges and misaligned chromosomes to be the major mitotic errors in the non-regressed tumours (Fig 1C). Interestingly, there was an increase in the percentage of chromosome bridges in the non-regressed tumours compared to the primary tumours, suggesting that non-regressed tumours acquire different mitotic errors. These results suggest that genomically stable tumours (K) are able to acquire a certain level of CIN during the course of acquiring therapeutic resistance, although initially unstable tumours (KM) have an increased chance of persisting during treatment.

## Sequencing of oncogene independent Kras and Kras/Mad2 tumours reveals recurrent SCNAs

To determine which alterations were involved in promoting oncogene independence, 7 K and 16 KM non-regressed tumours were subjected to low-pass whole-genome sequencing using an Illumina HiSeq platform, followed by somatic copy number alteration (SCNA) analysis (Fig 2A). Similar to what we found in primary tumours, genomes of KM non-regressed tumours were more frequently affected than K tumours. However, in line with the live imaging results, the mean of SCNAs was not significantly increased compared to K non-regressed tumours (Figs 2A and EV1A). Notably, some recurrent alterations in these tumours were a whole gain of chromosome 15 in 1.2% of K and 6.5% of KM non-regressed tumours (Fig EV1B and C) and a deletion, a partial chromosome loss and in 2 KM non-regressed tumours a gross rearrangement of chromosome 4. However, we found that the most frequent alteration in either cohort was a small ~2 megabase amplification on chromosome 6, present in 57% of K and 62.5% of KM nonregressed tumours. While whole gain of chromosome 15 and deletions and partial deletions of chromosome 4 were already present in some K and KM primary tumours (Fig EV1D and Rowald *et al*, 2016), amplifications in chromosome 6 were detected for the first time in the non-regressed tumours.

In an attempt to investigate the mechanism driving resistance, we focused on those tumours that did not re-express oncogenic Kras, as reactivation of the oncogenic driver was most probably driving resistance. Moreover, we reasoned that, if the chromosomal alterations were already present in the primary tumour, they might represent genomic events that cooperated with the oncogenic driver in facilitating mammary tumour formation but not involved in resistance. Finally, since amplification of chromosome 6 was the most frequently detected alteration, we focused on those tumours that presented this amplification. Read depth analysis revealed variation in both the size and copy number pattern of the amplicons, with some presenting concise duplications and others showing complex copy number alterations resembling more elaborate mechanisms such as breakagefusion-bridge cycles (Gisselsson *et al*, 2000; Fig EV2A). However, overlaying the amplifications across multiple regions revealed that all amplicons contained the Met oncogene (Fig 2B).

Met gene encodes for the receptor tyrosine kinase cMET, mainly expressed in cells of mesenchymal origin and implicated in the activation of proliferation, survival, motility and morphogenesis

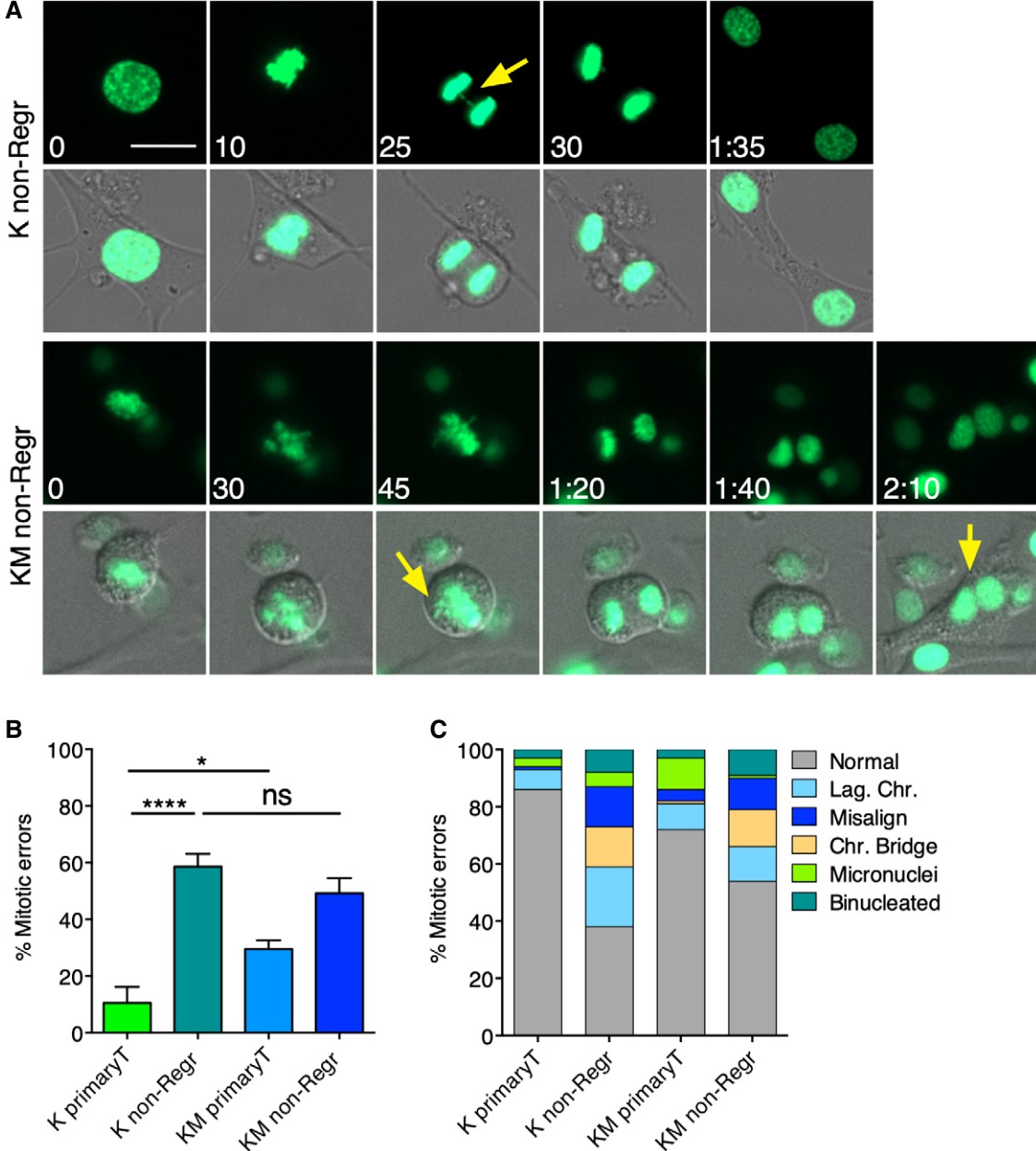

**Figure 1. Non-regressed K and KM tumours show high levels of chromosomal instability.**

A Representative micrographs of a time-lapse microscopy of K and KM non-regressed tumour cells (H2B-GFP green). Top: mitotic cell with a chromatin bridge (yellow arrow). Bottom: mitotic cell with misalignment and cytokinesis failure resulting in binucleation (yellow arrow).

B Percentage of mitotic errors in K and KM primary tumours and in K and KM non-regressed tumours. Ns: not significant; *$P < 0.01$, ****$P < 0.0001$; one-way ANOVA, Turkey's multiple comparison test. Mean ± SEM. Exact $P$ values are indicated in Appendix Table S1.

C Percentage of cells in K and KM primary tumours and in K and KM non-regressed tumours with the indicated mitotic errors. Scale bar 20 μm.

Data information: K primary ($n = 4$; 74 cells), KM primary ($n = 5$; 84 cells), K non-Regr ($n = 7$; 219 cells), KM non-Regr ($n = 5$; 247 cells).
Source data are available online for this figure.

signalling pathways (Birchmeier *et al*, 2003). Deregulation of cMET by gene amplification, overexpression or activating mutations (Tokunou *et al*, 2001; Lengyel *et al*, 2005) has been found in a variety of carcinomas, although the most common alteration is gene amplification and consequently protein overexpression and activation (Di Renzo *et al*, 1995). Analysis of cMet mRNA in K and KM non-regressed breast tumours (Fig EV2B) revealed a direct correlation between gene amplification and mRNA levels (Fig EV2C). Moreover, immunohistochemistry against phospho-cMet was positive in those tumours containing the chromosome 6 amplification and negative in the ones where the amplification was not present (Fig 2C and Table 1).

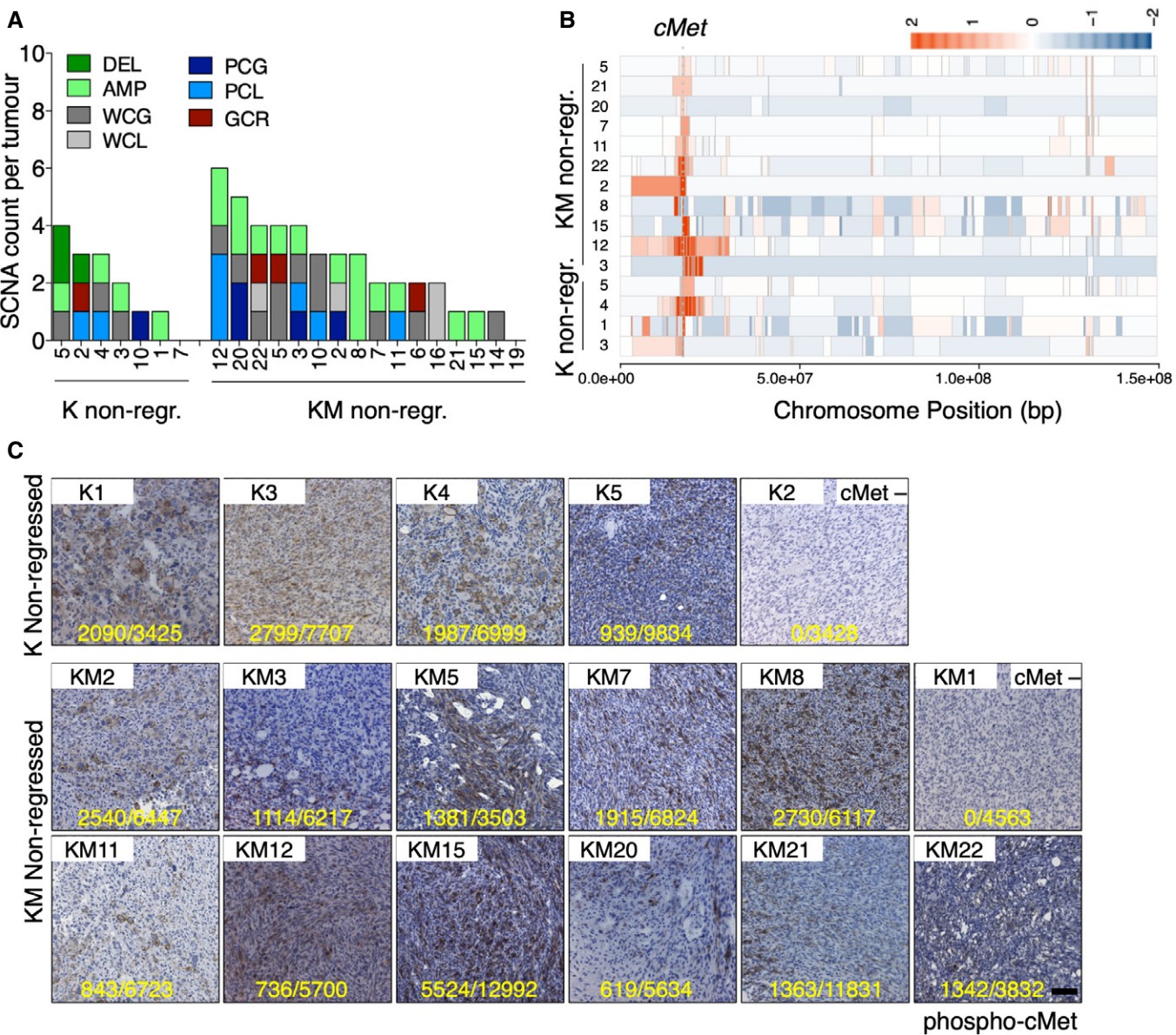

**Figure 2. Recurrent somatic copy number alterations in K and KM non-regressed tumours.**

A   Somatic copy number alterations in each individual Kras and Kras/Mad2 non-regressed tumours. 7 K non-regressed and 16 KM non-regressed tumours were sequenced. Focal deletion (DEL, green), focal amplification (AMP, light green), whole chromosome gain (WCG, dark grey) and loss (WCL, light grey), partial chromosome gain (PCG, dark blue) and loss (PCL, light blue) and gross chromosomal rearrangement (GCR, red).

B   Overlaying of the amplifications in chromosome 6 across multiple regions showing that all amplicons contained the cMet oncogene. Colour of the block corresponds to the segment mean (the degree to which a genome segment is lost (blue) or gained (red).

C   Immunostaining of phospho-cMet in non-regressed tumours carrying or not the amplification in chromosome 6. Yellow numbers indicate the total number of cMet-positive cells/total number of cells counted. Scale bar 100 μm. Numbers of the K and KM non-regressed tumours are described in Table 1.

Source data are available online for this figure.

## cMet amplification is not clonally dominant in primary tumours

To clarify whether the amplification on chromosome 6 was already present in the primary tumour, we first looked at the tumour evolution after doxycycline withdrawal and noticed that following a first period where tumours underwent a reduction in size, they continued to grow (Fig 3A). This suggests that in case

cMet amplification was not present in the primary tumour, it could have been acquired during this timeframe, forced by the selective pressure that oncogenic silence exerted within the tumour. We then looked if cMet-positive KM tumours resumed growth faster than K tumours. In fact, KM tumours partially regressed after doxycycline withdrawal and they needed an average of 38 days to grow back while K tumours took 133 days

**Table 1.  Non-regressed tumours used in this study.**

| Genotype | Name | Sequenced# SCNA | cMet ampl. by seq. | cMet upreg. QPCR Fig EV2B | cMet upreg. IHC Fig 2C | Kras re-expression[a] |
|---|---|---|---|---|---|---|
| Kras[G12D] | K1 | 1 | YES | YES | 61% | NO |
| Kras[G12D] | K3 | 2 | YES | YES | 36% | NO |
| Kras[G12D] | K4 | 3 | YES | YES | 28% | NO |
| Kras[G12D] | K5 | 4 | YES | YES | 9% | NO |
| Kras[G12D] | K6 | n.d. | n.d. | YES | n.d. | NO |
| Kras[G12D] | K2 | 3 | NO | NO | 0% | NO |
| Kras[G12D] | K7 | 0 | NO | NO | 0% | YES |
| Kras[G12D] | K8 | n.d. | NO | NO | n.d. | NO |
| Kras[G12D] | K9 | n.d. | NO | NO | n.d. | YES |
| Kras[G12D] | K10 | 1 | NO | NO | n.d. | NO |
| Kras[G12D] | K11 | n.d. | n.d. | n.d. | n.d. | n.d. |
| Kras[G12D]/Mad2 | KM2 | 3 | YES | YES | 39% | NO |
| Kras[G12D]/Mad2 | KM3 | 4 | YES | YES | 18% | NO |
| Kras[G12D]/Mad2 | KM4 | n.d. | n.d. | YES | n.d. | NO |
| Kras[G12D]/Mad2 | KM5 | 4 | YES | YES | 39% | NO |
| Kras[G12D]/Mad2 | KM7 | 2 | YES | YES | 28% | YES. Low |
| Kras[G12D]/Mad2 | KM8 | 3 | YES | YES | 44% | yes |
| Kras[G12D]/Mad2 | KM11 | 2 | YES | YES | 12% | NO |
| Kras[G12D]/Mad2 | KM12 | 6 | YES | YES | 13% | NO |
| Kras[G12D]/Mad2 | KM15 | 1 | YES | YES | 42% | NO |
| Kras[G12D]/Mad2 | KM20 | 5 | YES | YES | 11% | NO |
| Kras[G12D]/Mad2 | KM21 | 1 | YES | YES | 11% | NO |
| Kras[G12D]/Mad2 | KM22 | 4 | YES | YES | 35% | YES |
| Kras[G12D]/Mad2 | KM1 | n.d. | n.d. | NO | 0% | NO |
| Kras[G12D]/Mad2 | KM6 | 2 | NO | NO | 0% | NO |
| Kras[G12D]/Mad2 | KM9 | n.d. | n.d. | NO | n.d. | Yes |
| Kras[G12D]/Mad2 | KM10 | 3 | NO | NO | 0% | NO |
| Kras[G12D]/Mad2 | KM13 | n.d. | n.d. | NO | n.d. | NO |
| Kras[G12D]/Mad2 | KM14 | 1 | NO | NO | n.d. | NO |
| Kras[G12D]/Mad2 | KM16 | 2 | NO | NO | n.d. | NO |
| Kras[G12D]/Mad2 | KM17 | n.d. | n.d. | NO | n.d. | YES |
| Kras[G12D]/Mad2 | KM18 | n.d. | n.d. | NO | n.d. | YES |
| Kras[G12D]/Mad2 | KM19 | 0 | NO | NO | n.d. | YES Low |
| Kras[G12D]/Mad2 | KM23 | n.d. | n.d. | n.d. | n.d. | n.d. |

n.d: not done.
[a]Rowald *et al* (2016).

(Fig 3A), suggesting that already CIN tumours were more predispose to acquire this genetic modification.

Then, we determined the presence or absence of the cMet amplification in the primary tumours. We compared the low-coverage sequenced genomes of two primary tumours that were biopsied and their corresponding non-regressed tumours, after withdrawal of the oncogenic driver. As shown in Fig 3B, we found no detectable alterations in that specific region in the primary tumours. Additional immunohistochemistry staining of 3 biopsied primary tumours whose corresponding non-regressed tumour contained a cMet amplification showed negative phospho-cMet staining in the primary tumour and positive in the non-regressed (Fig 3C). Additionally, we analysed a panel of primary tumours at the time point just before doxycycline withdrawal. Phospho-cMet staining in 34 primary tumours was negative in all the tumour cells analysed (more than 400,000 cells in total; Fig EV3).

To further examine whether cMet amplification could be present in a small population of the tumours undetectable by whole-genome sequencing or immunohistochemistry, we resorted to analyse by digital PCR a panel of primary tumours at human endpoint (Fig 3D

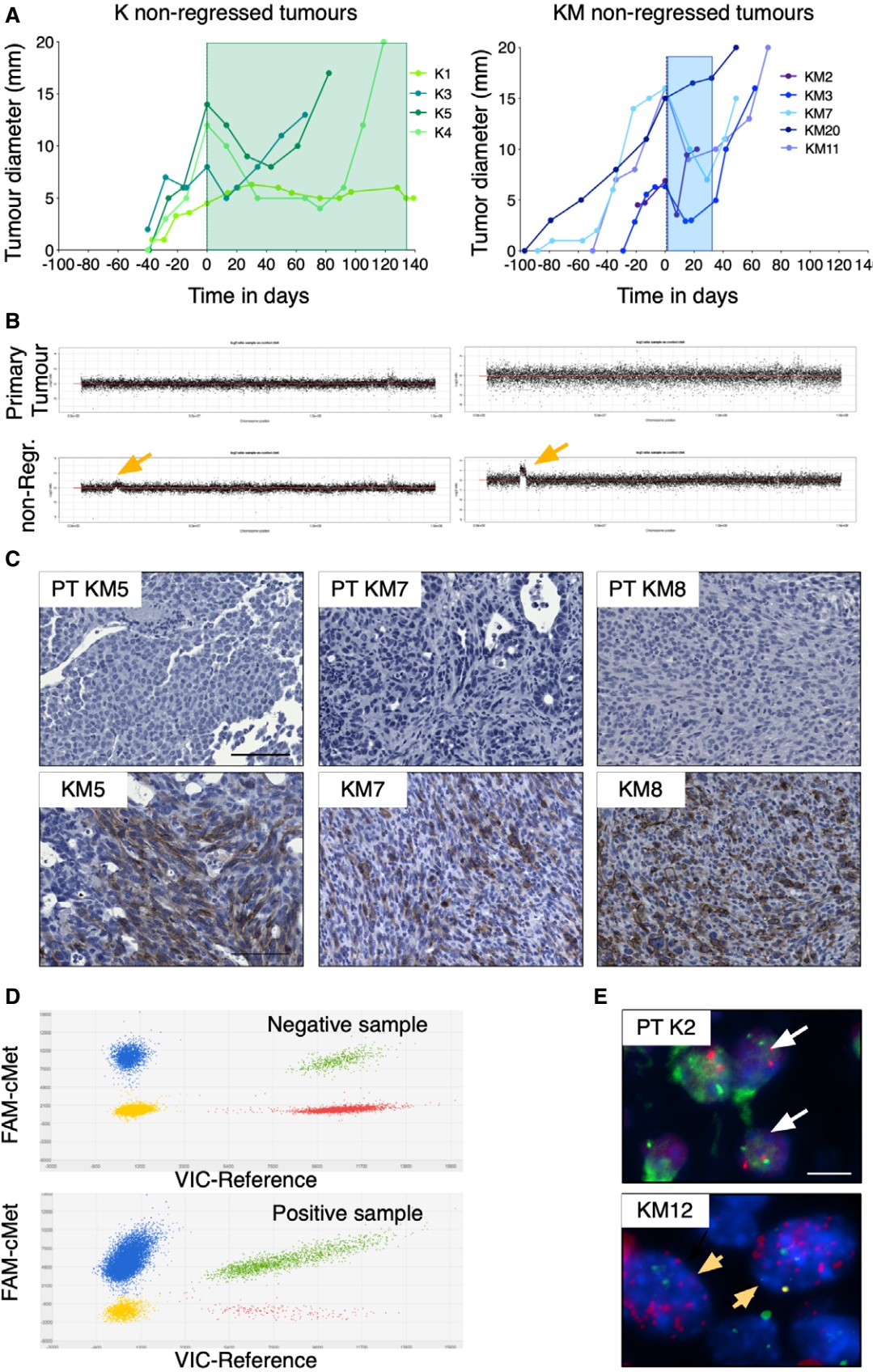

**Figure 3.**

**Figure 3. cMet amplification is not found in primary tumours.**

A  Tumour diameter before and after doxycycline withdrawal in 4 K and 5 KM breast tumours with cMet amplification. 0 indicates when doxycycline was removed. Each colour represents one tumour. Blue and green squares indicate the timeframe between doxycycline withdrawal and the moment in which tumours resumed growth.

B  Genome-wide $log_2$-ratio plots of chromosome 6 of two primary tumour biopsies and their corresponding non-regressed tumour showing no amplification in the primary tumour (upper panels) and a small amplification in the non-regressed tumours (bottom panels, yellow arrow).

C  Immunostaining of phospho-cMet in 3 biopsied primary tumours (PT) and their corresponding non-regressed tumours (KM5, KM7 and KM8, which are also shown in Fig 2C). Scale bar 100 μm.

D  Representative two-dimensional scatter plots constructed with overlaid dPCR data of the reference (VIC) and cMet (FAM) from one tumour without cMet amplification and one with cMet amplified. Dots represent results of independent PCRs in the wells of a digital PCR chip. Reactions in the bottom left corner (yellow) are negative for both targets, while the ones in the top right corner (green) are double positives. Reactions in the top left (blue) and bottom right (red) corners are positive for cMet and the reference targets, respectively.

E  Representative photographs of FISH staining with a probe for Met (red signal) and a probe for a reference gene EML4 (green signals) The upper panel is a negative example for cMet amplification containing 2 red and 2 green dots (white arrows). The lower panel shows an example of a tumour with cMet amplification (several red dots) and 2 green dots (yellow arrow). Scale bar 10 μm.

Source data are available online for this figure.

and Table 2). The measured ratio between cMet and a reference gene was never significantly different from 1 except for one primary tumour whose ratio was 1.5, a result explained by a whole chromosome 6 gain detected by whole-genome sequencing (PT KM15) (Table 2). Finally, we looked at the single-cell level for amplification of cMet by performing fluorescent *in situ* hybridization (FISH) with a probe against cMet. A probe recognizing EML4 gene was used as reference, given this gene is located in chromosome 17, which was almost never affected in the primary tumours. During the analysis, cells where 3 or more copies of cMet were accompanied by 3 or more copies of EML4 were discarded, since they might represent cells that underwent genome doubling. We found that the cMet: EML4 ratio on population level was close to 1 in all tumours, but on average 6.22% of cells in K and 5.67% of cells in KM tumours carried 3–5 copies of cMet (Fig 3E and Table 2). Although detection of cMet gene amplification by FISH has been previously considered positive when the number of copies exceed 5 in more than 15% of the nuclei (Cappuzzo *et al*, 2009), the FISH results were not sufficient to rule out the existence of cMet-positive cells. Therefore, we sought to test whether these primary tumour cells carrying more than three copies of cMet were indeed dependent on cMet by evaluating the effect of a cMet inhibitor on tumour growth. We first recapitulated the primary tumours by injecting tumour cells from K and KM tumours into the cleared fat pad of $Rag2^{-/-}$ immunocompromised animals (Fig 4A). When tumours reached 0.5 $cm^3$, we treated them with the cMet inhibitor tepotinib which has been proven to selectively inhibit the kinase activity of this protein, independently of its mechanism of activation (Bladt *et al*, 2013). We found no differences in tumour growth between control and treated mice (Fig 4B) and the results of FISH analysis of the treated tumours were similar to the untreated ones (Fig 4C). Moreover, the ratio treated/control obtained by dPCR was not significantly different from 1 in all tumours independently of the genotype (K or KM), suggesting that the presence of more than 2 copies of cMet in some of the cells did not lead to detectable difference in the sensitivity to the drug treatment in the tumour cell population.

Furthermore, we evaluated the presence of cMet amplification during regression, a few days after doxycycline withdrawal, since we considered it to be the most critical time point to claim an increase cMet activation after oncogene withdrawal. When primary tumours grown in $Rag2^{-/-}$ mice reached 1 $cm^3$, we removed doxycycline from the diet and monitored the tumours until they regressed to 0.5 $cm^3$,

giving them enough time for the possible acquisition of cMet amplification. FISH in these tumours (Fig 4D) showed a slight increase in the percentage of cells with more than 2 copies of cMet in few (1 K and 2 KM) tumours although the percentage of cells with more than five copies was close to 0%. Here again, dPCR measurement showed an equal ratio of cMet/reference in all tumours, suggesting that a significant cMet amplification did not occur during tumour regression.

Altogether, these results support the idea that *de novo* acquired chromosome alterations (such as the amplification of chromosome 6) could be responsible for driving the development of resistant subclones.

**Resistant tumours are dependent on cMet**

To determine whether non-regressed tumours depended on cMet expression to grow after withdrawal of the Kras initiating oncogene, we sought to determine whether they were sensitive to pharmacological cMet inhibition. We first recapitulated the non-regressed tumours by injecting non-regressed tumour cells from K and KM tumours into the cleared fat pad of $Rag2^{-/-}$ immunocompromised animals (Fig 5A), and once tumours reached an average of 0.5 $cm^3$, we treated the animals with the cMet inhibitor tepotinib. After 5-day treatment, tepotinib strongly reduced tumour growth in cMet-positive xenografts compared to a vehicle control, whereas no significant differences in growth were observed in tumours without cMet amplification between drug and vehicle group (Fig 5B). To further confirm the therapeutic ability of tepotinib, the cMet-positive xenografts were treated for 2 additional weeks. As shown in Fig 5B, after a first period in which tumours were sensitive to the drug, they resumed growth, suggesting that they became resistant to cMet inhibition.

To confirm that cMet phosphorylation was inhibited upon short exposure to tepotinib, we analysed the phosphorylation status of cMet in these xenografts. Tumours carrying cMet amplification were highly positive for phospho-cMet after vehicle control treatment, while 5-day treatment with tepotinib impaired the *in vivo* phosphorylation of cMet (Fig 5C). As expected, cMet remained unphosphorylated upon vehicle treatment in non-amplified cMet tumours. Altogether, these results suggest that K and KM resistant tumours that amplified cMet are dependent on cMet for survival as treatment with a specific cMet inhibitor hampers tumour growth.

Tepotinib binds specifically to cMet protein and impairs its activation and function, leading to cell cycle arrest and apoptosis (Bladt

**Table 2. Primary tumours used in this study.**

| Genotype | Name | FISH MET: EML4 | | dPCR Ratio | % p-cMET[+] |
|---|---|---|---|---|---|
| | | Ratio | %cells ≥3 copies | | |
| Kras$^{G12D}$ | PT K1 | 1.01 | 4.80 | 1.01 | 0/11313 |
| Kras$^{G12D}$ | PT K2 | 1.02 | 4.55 | 1.02 | 0/7555 |
| Kras$^{G12D}$ | PT K3 | 0.97 | 2.20 | 0.98 | 0/5307 |
| Kras$^{G12D}$ | PT K4 | 1.01 | 5.05 | 0.99 | 0/10199 |
| Kras$^{G12D}$ | PT K5 | | | 0.98 | 0/22759 |
| Kras$^{G12D}$ | PT K6 | 0.96 | 0.02 | 1.00 | 0/12067 |
| Kras$^{G12D}$ | PT K7 | | | 1.07 | 0/17125 |
| Kras$^{G12D}$ | PT K8 | 0.97 | 6.41 | 0.95 | 0/16602 |
| Kras$^{G12D}$ | PT K9 | | | 1.06 | 0/11755 |
| Kras$^{G12D}$ | PT K10 | | | 1.01 | 0/16521 |
| Kras$^{G12D}$ | PT K11 | 1.01 | 4.12 | 1.05 | 0/7591 |
| Kras$^{G12D}$ | PT K12 | | | 1.03 | 0/14508 |
| Kras$^{G12D}$ | PT K13 | 0.95 | 5.77 | 0.87 | 0/13809 |
| Kras$^{G12D}$ | PT K14 | 1.02 | 14.50 | 0.93 | 0/9927 |
| Kras$^{G12D}$ | PT K15[a] | | | 1.41 | 0/11764 |
| Kras$^{G12D}$ | PT K16 | | | 0.97 | 0/12892 |
| Kras$^{G12D}$ | PT K17 | | | | 0/10418 |
| Kras$^{G12D}$ | PT K18 | | | | 0/9657 |
| Kras$^{G12D}$ | PT K19 | | | | 0/13295 |
| Kras$^{G12D}$ | PT K20 | | | | 0/8469 |
| Kras$^{G12D}$ | PT K21 | | | | 0/8059 |
| Kras$^{G12D}$ | PT K22 | | | | 0/12054 |
| Kras$^{G12D}$ | PT K23 | | | | 0/9037 |
| Kras$^{G12D}$ | PT K24 | 1.43 | 21.00 | 0.99 | 0/12872 |
| Kras$^{G12D}$ | PT K25 | 0.97 | 3.30 | 0.83 | 0/11299 |
| Kras$^{G12D}$ | PT K26 | 1.01 | 3.00 | 1.02 | |
| Kras$^{G12D}$ | PT K27 | | | 1.06 | |
| Kras$^{G12D}$ | PT K28 | | | 1.01 | |
| Kras$^{G12D}$ | PT K29 | | | 0.98 | |
| Kras$^{G12D}$ | PT K30 | | | 0.96 | |
| Kras$^{G12D}$ | PT K31 | | | 0.96 | |
| Kras$^{G12D}$ | PT K32 | | | 0.97 | |
| Kras$^{G12D}$ | PT K33 | | | 1.00 | |
| Kras$^{G12D}$/Mad2 | PT KM1 | 1.008 | 3.51 | 0.92 | 0/6615 |
| Kras$^{G12D}$/Mad2 | PT KM2 | 1.004 | 1.83 | 0.89 | 0/14488 |
| Kras$^{G12D}$/Mad2 | PT KM3 | | | 0.98 | 0/14986 |
| Kras$^{G12D}$/Mad2 | PT KM4[a] | | | 1.35 | 0/16605 |
| Kras$^{G12D}$/Mad2 | PT KM5 | 0.99 | 5.66 | 1.04 | 0/5409 |
| Kras$^{G12D}$/Mad2 | PT KM6 | | | | 0/8293 |
| Kras$^{G12D}$/Mad2 | PT KM7 | | | 1.00 | 0/4819 |
| Kras$^{G12D}$/Mad2 | PT KM8 | 1.009 | 2.98 | | 0/6987 |
| Kras$^{G12D}$/Mad2 | PT KM9 | | | | 0/12622 |
| Kras$^{G12D}$/Mad2 | PT KM10 | | | | 0/10461 |

**Table 2** (continued)

| Genotype | Name | FISH MET: EML4 | | dPCR Ratio | % p-cMET[+] |
|---|---|---|---|---|---|
| | | Ratio | %cells ≥3 copies | | |
| Kras$^{G12D}$/Mad2 | PT KM11 | 0.96 | 6.9 | | 0/13470 |
| Kras$^{G12D}$/Mad2 | PT KM12 | 1.02 | 13.15 | 1.02 | 0/6478 |
| Kras$^{G12D}$/Mad2 | PT KM13 | | | 1.00 | |
| Kras$^{G12D}$/Mad2 | PT KM14 | | | 0.48 | |
| Kras$^{G12D}$/Mad2 | PT KM15[b] | 2.47 | 40.5 | 1.50 | |
| Kras$^{G12D}$/Mad2 | PT KM16 | | | 0.93 | |
| Kras$^{G12D}$/Mad2 | PT KM17 | | | 1.07 | |
| Kras$^{G12D}$/Mad2 | PT KM18 | | | 0.85 | |
| Kras$^{G12D}$/Mad2 | PT KM19 | | | 1.38 | |
| Kras$^{G12D}$/Mad2 | PT KM20 | | | 0.97 | |
| Kras$^{G12D}$/Mad2 | PT KM21 | | | 0.97 | |

[a]Whole chromosome 16 loss (location of the reference gene used in the dPCR).
[b]Whole chromosome 6 gain.

et al, 2013). To confirm that tumour reduction in cMet\-positive xenografts was a consequence of these two mechanisms, we quantified tumour cell proliferation (pH3[+]) and cell death (cleaved caspase 3[+]) (Figs 5D–E and EV4). Strikingly, cMet-positive tumours showed higher percentage of proliferating cells compared to cMet-negative tumours (26.75% versus 9.33%, $P < 0.001$; Figs 5D and EV4), while treatment with tepotinib in cMet-positive tumours led to a significant reduction in pH3, showing equivalent levels to the cMet-negative tumours. Importantly, cMet inhibition did not impinge on cell proliferation of cMet-negative tumours, highlighting the specific effect of this inhibitor. Similarly, we found increased cell death in the mammary tissue of cMet-positive tumours compared to the rest of groups in the study (Figs 5E and EV4).

Altogether, these results confirm that all K and KM non-regressed tumours that presented an amplification on chromosome 6 are addicted to cMet, since treatment with tepotinib led to tumour regression by decreasing proliferation and inducing apoptosis, confirming that cMet is essential for the proliferation and/or viability of tumour cells.

# Discussion

Tumour resistance and recurrence remain to date one of main causes of breast cancer-related deaths. Underlying resistance is the principal of intra-tumour heterogeneity, which expedites the generation of more malignant cancer cells or clones resistant to therapeutic intervention (McGranahan & Swanton, 2017). Indeed, the DNA index, a proxy for karyotypic heterogeneity, correlates with important clinical features including tumour size, grade, lymph node metastasis and ER status (Dayal et al, 2013). Here, we present results that mechanistically demonstrate how, in the context of breast tumorigenesis and oncogene withdrawal, CIN allows tumour cell populations to evolve past their dependence on their initiating oncogene via the production of oncogenic SCNAs (e.g. Met amplification).

Increasing number of studies have reported clonal mutations as a mechanism to resist both targeted and chemotherapy in many cancer types (Redmond et al, 2015). Nevertheless, treatment resistance can occur in tumours showing high levels of SCNA and CIN. In colorectal cancer, for instance, karyotypic heterogeneity might be responsible for drug resistance compared to karyotypically stable tumours (Lee et al, 2011) and also associated with worse prognosis (Walther et al, 2008). Moreover, Kwong and colleagues found that in an environment with strong selective pressure such as therapy, unstable tumours develop resistance by selecting for recurrent aneuploidies (Kwong et al, 2017). Supporting this idea, we found that upon doxycycline withdrawal, higher percentage of chromosomically unstable KM tumours were able to resist and continue growing compared to K tumours, indicating that the presence of CIN during primary tumour development increases the chance to develop therapy resistance in a model of Kras-driven breast cancer (Rowald et al, 2016). Surprisingly, sequencing data from resistant tumours showed that although the percentage of genomic alterations per chromosome was higher in KM non-regressed tumours, the total number of SCNAs in K and KM was similar (Figs 2 and EV1). Moreover, genomes of both K and KM resistant non-regressing tumours were highly unstable with tumour cells frequently generating mitotic errors (Fig 1), suggesting that acquisition of a CIN phenotype might be a prerequisite for the development of therapy resistance. We found no correlation between the number of SCNAs present in the resistant tumours and the frequency of mitotic errors seen by video microscopy, probably due to the fact that resistant tumour cells are highly CIN and constantly evolving. We observed an increase in binucleated cells in non-regressed compared to primary tumours, and although it will be tempting to speculate that tetraploidy could partly account for the increase in resistance in KM tumours, also an increase in chromosome bridges, misalignment and lagging chromosomes was observed. Interestingly, we noticed an increased number of chromosome bridges in the non-regressed tumours from both K and KM genotypes compared to the primary tumours and further genomic analyses inferred that the complex copy number alterations resembled a mechanism of breakage-fusion-bridge cycle. The fact that in primary tumours of either K or KM genotypes, chromosome bridges were very rarely observed

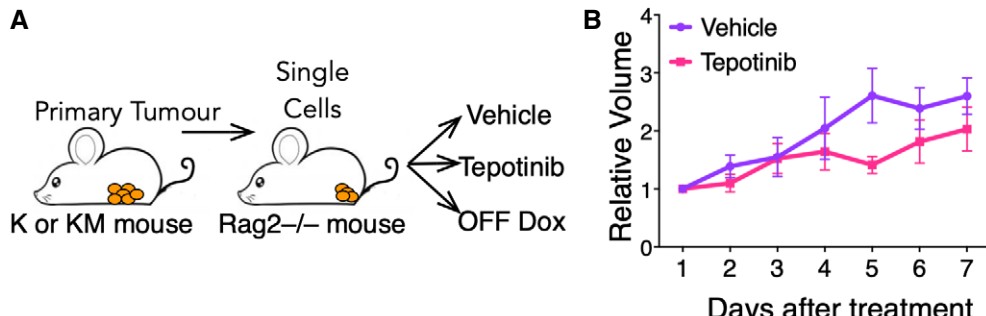

**A**

**B**

**C**

| Genotype | Name | Treatment | FISH MET:EML4 | | dPCR |
|---|---|---|---|---|---|
| | | | RATIO TREAT:CTL | %cells ≥3 copies of cMet | RATIO TREAT:CTL |
| Kras$^{G12D}$ | RK2 | CTL | n.d. | n.d. | |
| | RK2 | TEPOTINIB | n.d. | n.d. | 0,90 |
| | RK3 | CTL | | 7,60 | |
| | RK3 | TEPOTINIB | 1,18 | 7,90 | 1,16 |
| | RK4 | CTL | | 4,50 | |
| | RK4 | TEPOTINIB | 1,10 | 7,90 | 0,89 |
| Kras$^{G12D}$/Mad2 | RKM1 | CTL | | 5,80 | |
| | RKM1 | TEPOTINIB | 2,25 | 15,40 | 0,88 |
| | RKM2 | CTL | | 10,52 | |
| | RKM2 | TEPOTINIB | 0,57 | 5,15 | 0,95 |
| | RKM3 | CTL | | 3,75 | |
| | RKM3 | TEPOTINIB | 1,17 | 1,07 | 0,95 |
| | RKM4 | CTL | | 6,60 | |
| | RKM4 | TEPOTINIB | 1,05 | 5,00 | 1,03 |

**D**

| Genotype | Name | Treatment | FISH MET:EML4 | | dPCR |
|---|---|---|---|---|---|
| | | | RATIO TREAT:CTL | %cells ≥3 copies of cMet | RATIO TREAT:CTL |
| Kras$^{G12D}$ | RK1 | CTL | | 11,00 | |
| | RK1 | OFF DOX | 0,78 | 3,06 | 0,98 |
| | RK2 | CTL | n.d. | n.d. | |
| | RK2 | OFF DOX | n.d. | 6,25 | 0,94 |
| | RK4 | CTL | | 4,50 | |
| | RK4 | OFF DOX | 1,24 | 10,00 | 0,96 |
| Kras$^{G12D}$/Mad2 | RKM1 | CTL | | 5,80 | |
| | RKM1 | OFF DOX | 1,25 | 8,00 | 0,93 |
| | RKM2 | CTL | | 10,52 | |
| | RKM2 | OFF DOX | 0,83 | 3,80 | 1,03 |
| | RKM3 | CTL | | 3,75 | |
| | RKM3 | OFF DOX | 1,63 | 8,08 | 1,01 |

Figure 4.

**Figure 4. cMet amplification is not clonally dominant in primary tumours.**

A Schematic of the experiment. K and KM animals were set on doxycycline food until mammary tumours developed. Primary tumours were collected and single cells injected into Rag2$^{-/-}$ animals to recapitulate the tumours followed by either no treatment, drug treatment or switched to a normal diet.

B Relative volume of tumours grown in 12 Rag2$^{-/-}$ animals after treatment with tepotinib or vehicle control (3K and 3KM tumours for each condition). No statistical significance was found by one-way ANOVA. Mean ± SEM. Exact *P* values are indicated in Appendix Table S2.

C Quantification of cMet copy number detected by digital PCR or FISH in primary tumours injected into Rag2$^{-/-}$ and treated with vehicle control or tepotinib.

D Quantification of cMet copy number detected by digital PCR or FISH in primary tumours injected into Rag2$^{-/-}$ and fed with doxycycline or after doxycycline withdrawal.

Source data are available online for this figure.

indicates that the breakage-fusion-bridge cycle (Gisselsson *et al*, 2000) could have occurred after doxycycline withdrawal. Although it will be difficult to directly test whether chromosome 6 is the one present in those bridges, it is important to note that gain of chromosome 6 has already been observed in mouse models of Ras- or EGFR-driven lung cancer (McFadden *et al*, 2016).

MET alterations (mutations, deletions and amplifications) have been found in various human cancers (cBioPortal). Pan-cancer analysis (Zehir *et al*, 2017) reveals that primary tumours with MET amplification have higher overall survival rate (11.83 months) compared to metastatic tumours (9.88 months). However, MET amplification appears at a very low percentage in different breast cancer data sets both in primary (TCGA BRCA (0.4%), (Pereira *et al*, 2016) (1.5%), (Razavi *et al*, 2018) (0.1%)) and metastatic tumours (2.3%) (Lefebvre *et al*, 2016), MBC project 2018 (1.3%), (0.2%) (Zehir *et al*, 2017), (0.15%) (Razavi *et al*, 2018). Future studies addressing the relationship between aneuploidy and MET under different oncogenic drivers could underline the importance of MET inhibitors as prognostic markers in the treatment of this disease.

The exact timing of the cMet amplification appearance remains elusive. There are evidences that suggest cMET amplification can be acquired *de novo*, induced by the treatment (Engelman *et al*, 2008). In our model, cMet amplification drives resistance in half of the tumours and confers a proliferative advantage compared to other genomic alterations present in the rest of the non-regressed tumours. However, this amplification was not detected in any of the primary tumours analysed by whole-genome sequencing, consistent with a model of acquired therapy resistance, such as the one described for glioblastoma (Kim *et al*, 2015). The absence of the cMet amplicon by DNA sequencing in biopsies of tumours whose corresponding non-regressed tumour showed cMet amplification suggests that either the cells were present in very small numbers or that the amplification occurred after oncogene withdrawal. The use

of more sensitive techniques such as dPCR and FISH provided further evidences against the possibility that this alteration was already present in the primary tumour and remained undetected due to insufficient sequencing depth. The ratio cMet:reference gene was not significantly different from 1 in 39 and 18 primary tumours analysed by dPCR and FISH, respectively, strengthening the idea that cMet amplification was acquired as a mechanism to overcome the selective pressure exerted by oncogene withdrawal.

In a recent study in triple-negative breast cancer (TNBC), the authors found that the majority of the mutations present in therapy resistant patients were already present during pre-treatment stages, although with lower frequencies after treatment. In addition, resistant patients showed higher levels of aneuploidy pre-treatment than patients that responded to chemotherapy (Kim *et al*, 2018). In our model, therapy resistance arises more frequently in aneuploid primary tumours, very much in line with these observations. This is supported by the fact that all K and KM non-regressed tumours carrying cMet amplification resumed growing in the presence of tepotinib. Moreover, some alterations present in primary tumours such as whole gain of chromosome 15 and gross rearrangement of chromosome 4 showed even higher frequencies after treatment, suggesting that more unstable genotypes were positively selected during the course of the treatment.

In summary, we have shown that chromosomal instability acts as a source of genetic variability that under strong selective pressure, such as during targeted therapy, confers tumour cells with an evolutionary advantage. We further demonstrate that the few initially chromosomally stable cancers that manage to persist during treatment do so concomitantly with the acquisition of CIN and this increases the chances of developing secondary resistances. Future experiments using CIN mouse models together with different oncogenic drivers will be useful to advance our understanding of resistant mechanisms to targeted therapies.

**Figure 5. cMet amplified non-regressed tumours respond to cMet treatment.**

A Schematic of the experiment. K and KM animals were set on doxycycline food until mammary tumours developed. Then, switched to a normal diet and non-regressed tumours collected. Single cells were then injected into Rag2$^{-/-}$ animals to recapitulate the non-regressed tumours followed by drug treatment or vehicle control treatment.

B Relative volume of tumours grown in Rag2$^{-/-}$ animals after treatment with tepotinib or vehicle control. One-way ANOVA, Tukey's multiple comparison. Ns: not significant; **$P < 0.024$. Mean ± SEM. A total of 19 Rag2$^{-/-}$ mice injected with cMet expressing tumours were treated with tepotinib and 14 used as controls. For tumours with no cMet, 13 Rag2$^{-/-}$ mice were treated with tepotinib and 9 were used as control. Exact *P* values are indicated in Appendix Table S3.

C Immunostaining of phospho-cMet in the same tumours grown in Rag2$^{-/-}$ mice after treatment with tepotinib or vehicle control. Scale bar 200 μm.

D Quantification of pH3 in tumours grown in Rag2$^{-/-}$ animals after 5 days treatment with the cMet inhibitor tepotinib or vehicle control.

E Quantification of caspase 3 in tumours grown in Rag2$^{-/-}$ animals after 5 days treatment with the cMet inhibitor tepotinib or vehicle control.

Data information: (D, E) One-way ANOVA, Tukey's multiple comparison; **$P < 0.005$; ****$P < 0.0001$. Mean ± SEM. At least four tumours per condition were analysed, and a minimum of 35 fields of view (FOV) or 12,000 cells were counted for each condition. Exact *P* values are indicated in Appendix Tables S3 and S4, respectively.
Source data are available online for this figure.

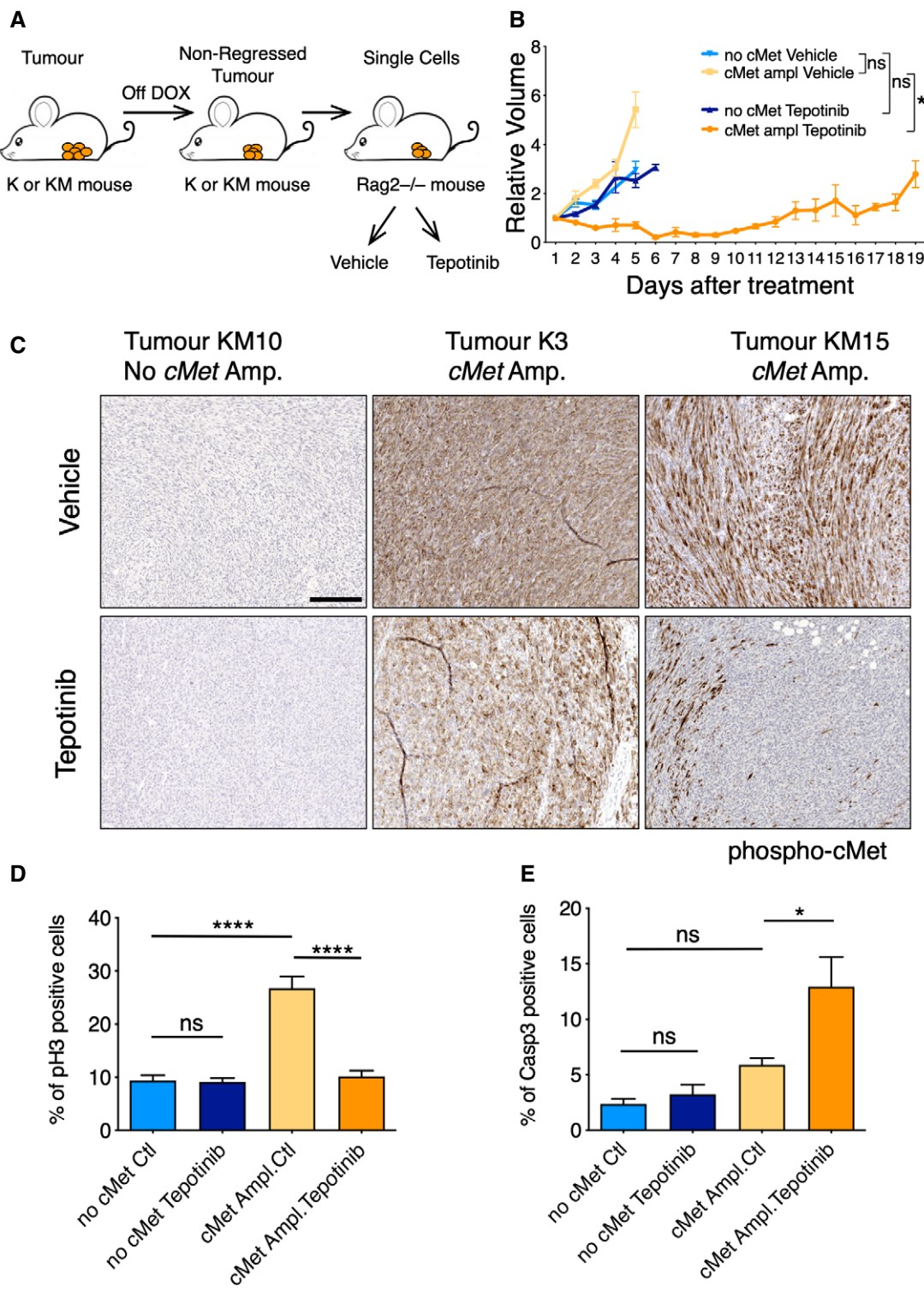

**Figure 5.**

## Materials and Methods

### Mouse models

All animals were in a FVB background and generated as described in Rowald et al (2016). Mad2 transgenic mice were the traditional TetO-Mad2 (Sotillo et al, 2007) and TetO-Kras[G12D] were described in Fisher et al (2001). All mice were housed in specific pathogen-free conditions, and breeding and experimentation were performed at the EMBL and DKFZ animal facilities, in accordance with institutional guidelines, and were approved by the Regierungspräsidium Karlsruhe, Germany, under permit number G231/15. For transgene

induction, 8-week-old females were treated with doxycycline, administrated via impregnated food pellets (625 mg/kg; Harlan-Teklad). When tumour size reached 1 cm$^2$, doxycycline food was replaced by normal food. Tumour growth was monitored regularly, and animals were sacrificed when their size reached 1.5 cm$^3$. Rag xenografts: 250,000 tumour cells per sample were injected into the cleared fat pad of 21-day-old Rag2$^{-/-}$ animals under isoflurane inhalation (2.5% in 0.8 l/min, Esteve) and in accordance with local disinfection and sterilization guidelines. Once tumours reached a volume of 0.5 cm$^2$, we treated the animals daily by intraperitoneal injection with either tepotinib (EMD1214063) (30 mg/kg) or vehicle (Solutol) (Bladt *et al*, 2013). Tumour size [length (L) and width (W)] was measured every 2 days, and the tumour volume was calculated using the formula $L \times W^2/2$.

### Tumour cell culture

Harvested tumours were digested with 150 U/ml Collagenase type 3 (Worthington, CLS3) and 20 mg/ml Liberase Blendzyme 2 (Roche, 11988425001) for 1 h, washed with PBS and dissociated with 0.25% Trypsin (Life Technologies, 25200056). Cells (not tested for mycoplasma) were cultured on 6-well plates (Corning) in serum-free mammary epithelial basal medium (MEBM) with supplements (Lonza, CC-3150). To generate H2B-GFP expressing mammary tumours, cells were infected with a lentivirus carrying H2B-GFP. For time-lapse imaging, cells were cultured on 8-well chambered cover glass (Thermo Scientific, 155411) and imaging during 15 h was performed on a Zeiss Cell Observer with 2-μm optical sectioning across 18-μm stack, 12 frames/h. Zeiss Zen 2 software served for image analysis.

### Immunodetection in tissue sections

Immunohistochemistry was performed using formalin-fixed paraffin-embedded sections. Following deparaffinization with xylene and rehydration with graded ethanol, antigen retrieval was performed using 0.09% (v/v) unmasking solution (Vector Labs) for 30 min in a steamer. Inactivation of endogenous peroxidases was carried out using 3% hydrogen peroxide (Sigma) for 10 min. Secondary antibody staining and biotin–streptavidin incubation were performed using species-specific VECTASTAIN Elite ABC Kits (Vector Labs). DAB Peroxidase Substrate Kit (Vector Labs) was utilized for antibody detection. Primary antibodies used were anti-pH3 Ser10 (1:200, Cell Signalling, 9701), cleaved caspase 3 (1:200, Cell Signalling, 9661) and Phospho-Met (Tyr1234/1235) (1:300, Cell Signaling, 3077). Tumour sections were visualized under a TissueFAXS slide scanning platform (TissueGnostics, Vienna, Austria). For the pH3 and Casp3, the quantitation was performed using StrataQuest software (TissueGnostics) to determine the percentage of pH3$^+$ or Casp3$^+$ cells.

### FISH

Interphase FISH was performed on formalin-fixed paraffin-embedded 5-μm sections. Following deparaffinization with xylene and rehydration with graded ethanol, antigen retrieval was performed using 0.09% (v/v) unmasking solution (Vector Labs) for 30 min in a steamer. Tissues were digested with 0.005% Pepsin at 37°C for 15 min, washed with 2× saline/sodium citrate for 3 min and dehydrated in an ethanol series (70%, 85%, 100%) for

### The paper explained

**Problem**

Therapy resistance is one of the main causes of death in breast cancer patients. During the course of treatment, acquisition of new genetic alterations can favour oncogene independence and therefore progression of the disease. Chromosome instability (CIN) acts as a powerful source of variability, compromising the efficacy of targeted therapy.

**Results**

Using a Kras-driven breast cancer mouse model, we showed that genomically unstable tumours have high chances of persisting during treatment. Furthermore, initially stable tumours are also able to acquire CIN as a resistance mechanism.

**Impact**

This study points out the important role of CIN in acquisition of therapy resistance and highlights the necessity to combine current treatments with drugs that specifically impair progression of unstable cells.

3 min each. Hybridization was performed using Abbott Molecular Thermobrite system with the following programme: denaturation 76°C for 5 min, hybridization at 37°C for 20 h. Posthybridization, washes were performed with 0.4× saline/sodium citrate/0.1% Tween20 at RT for 2 min; 0.4× saline/sodium citrate/0.1% Tween20 at 74°C for 2 min; cooled in 2× saline/sodium citrate/0.1% Tween20 for 2 min; and 2× saline/sodium citrate for 3 min. Finally, slides were mounted with Prolong Diamond (Life Technologies, P36966).

Fluorescent *in situ* hybridization probes were prepared from purified BAC clones RP23 clone 416H6 (for cMet gene located within A2 of chromosome 6) and RP23 clone 193B15 (for EML4 gene located in chromosome 17; Maddalo *et al*, 2014) labelled with Red 650-dUTP (Enzo) and SpectrumOrange-dUTP (Vysis), respectively, by nick translation according to standard procedures. Signal for hybridization for each probe was checked in a minimum of 70 interphase cells in each tumour sample. FISH samples were imaged using Zeiss Cell Observer microscope in the DKFZ Light Microscopy Facility. Images were analysed using FIJI software.

### RNA work

Snap-frozen tissue was grinded with mortar and pestle on dry ice. For RNA extraction, 30 mg of tissue was used. All further steps were performed via the RNeasy Mini Kit (Qiagen) according to technical specifications. For cDNA synthesis, we followed the specifications of the QuantiTect Reverse Transcription Kit (Qiagen). Real-time quantification was performed on a starting material of 8 ng cDNA with SYBR Green PCR Master Mix (2×) (Applied Biosystems) in a LightCycler II® 480 (Roche). Primers used were cMet F: CATTTT TACGGACCCAACCA and cMet R: TGTCCGATACTCGTCACTGC and actin F: GCTTCTTTGCAGCTCCTTCGT and actin R: ACCAGCCGCA GCGATATCG.

### DNA sequencing and analysis

Genomic DNA was extracted from mouse tumour cells using the DNA Blood Mini Kit (QIAGEN). Library preparation and

low-coverage sequencing (3×) were pursued on an Illumina HiSeq 2500 platform (Illumina) using 50-base pair single-end reads as described previously (Rowald *et al*, 2016). Reads were aligned to the mm10 build of the mouse reference genome using Burrows-Wheeler Aligner (BWA; version 0.7.10; Li & Durbin, 2009). Coverage files were calculated and $\log_2$-normalized to mouse genomic DNA derived from normal mammary tissue. Circular binary segmentation (CBS; R package) was applied, and somatic copy number alterations were categorized as follows. Whole chromosome gains/losses were defined as chromosome-wide shifts in the segmentation of a chromosome, whereas partial chromosome gains/losses entailed changes spanning at least one-fifth of the chromosome. Focal amplifications and deletion encompassed events smaller than this. When the number of copy number state switches on a chromosome exceeded ten, we called them gross chromosomal rearrangements.

### Digital PCR

Genomic DNA was extracted from mouse tumour cells, and DNA samples were dispensed in the chip-based QuantStudio™ 3D Digital PCR System (Thermo Fisher Scientific) and amplified using selected TaqMan™ CNV Assays for mouse cMET gene (Mm00193012_cn). Mouse TFRC or TERT was used as reference gene (4458367 and 4458368, Thermo Fisher Scientific). dPCRs were set up in a final volume of 14.5 µl, containing 7.5 µl of 2× QuantStudio™ 3D Digital PCR Master Mix, 0.725 µl of each TaqMan® probe (Life Technologies) and 15 ng of gDNA templates adjusted in 5.8 µl water. Thermal cycling was as follows: 10 min at 96°C, 39 cycles at 60°C for 2 min, 30 s at 98°C and a final elongation step of 2 min at 60°C. Subsequent analysis and post-processing were performed with the QuantStudio 3D AnalysisSuite Software.

### Statistical analysis

Statistical analysis was carried out using Prism6 (GraphPad). $P$ values were as follows: $*P < 0.05$, $**P < 0.01$, $***P < 0.001$, $****P < 0.0001$. Scatterplots show mean and SEM. Box-and-whisker plots show median interquartile ranges plus minimum to maximum range. The number of animals is represented with $n$.

## Data availability

The accession number for the sequencing data reported in this paper is ENA: PRJEB23645 (https://www.ebi.ac.uk/ena/browser/view/PRJEB23645).

**Expanded View** for this article is available online.

## Acknowledgements

We thank Simone Kraut, Jessica Steiner, the DKFZ light microscopy unit and the DKFZ mouse facility for excellent technical assistance. We also thank members of the Sültmann lab for the technical help in dPCR, Andrea Ventura for the EML4 BAC for FISH and members of the Sotillo lab for critical reading of the manuscript. L.S is supported by a postdoctoral fellowship from Fundacion Ramon Areces. Work in the R.S laboratory was supported by an ERC starting grant (#281614) and the Howard Hughes Medical Institute.

## Author contributions

KR generated the non-regressed tumours. LS and KR characterized the non-regressed tumours. LS analysed the phenotype *in vivo* and performed mouse treatments. CB and JOK performed the sequencing analysis. KS performed the dPCR experiments. SK helped with the discussion. RS designed and supervised the study and analysed data. RS and LS wrote the manuscript with the help of CB.

## Conflict of interest

The authors declare that they have no conflict of interest.

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
