## [Review Process File · EMBO Molecular Medicine]

Acquisition of chromosome instability is a mechanism to evade oncogene addiction

Lorena Salgueiro, Christopher Buccitelli, Konstantina Rowald, Kalman Somogyi, Sridhar Kandala, Jan O. Korbelt and Rocio Sotillo

Review timeline:	Submission date:	28 May 2019
	Editorial Decision:	19 June 2019
	Revision received:	7 December 2019
	Editorial Decision:	20 December 2019
	Revision received:	4 January 2020
	Accepted:	16 January 2020

Editor: Lise Roth

Transaction Report:

1st Editorial Decision

19 June 2019

Thank you for the submission of your manuscript to EMBO Molecular Medicine. We have now received feedback from the three reviewers who agreed to evaluate your manuscript. As you will see from the reports below, the referees acknowledge the interest of the study. However, they also raise substantial concerns on your work, which should be convincingly addressed in a major revision of the present manuscript. In particular, it will be important to address the timeline of cMET modification occurrence, and to increase the clinical relevance of the xenograft experiments. Moreover, attention should be given to placing the study in the context of publicly available data.

Addressing the reviewers' concerns in full will be necessary for further considering the manuscript in our journal, and acceptance of the manuscript will entail a second round of review. EMBO Molecular Medicine encourages a single round of revision only and therefore, acceptance or rejection of the manuscript will depend on the completeness of your responses included in the next, final version of the manuscript. For this reason, and to save you from any frustrations in the end, I would strongly advise against returning an incomplete revision.

I look forward to receiving your revised manuscript.

***** Reviewer's comments *****

Referee #1 (Remarks for Author):

In this manuscript, Salgueiro et al. use an elegant genetic model to study the effects of chromosomal instability on the ability of tumors to persist after oncogene withdrawal. Interestingly, they found that instability promoted tumor oncogene independence through the amplification of cMET. They then use a cMET inhibitor and found that non regressed tumors that amplified cMET were selectively sensitive to cMET inhibitors. Overall, this manuscript studies an important question and convincingly shows the importance of chromosomal instability in promoting resistance and explores a specific vulnerability that can be subsequently used to selectively kill adapted tumors. I support the publication of this manuscript after the authors have addressed the following points:

- It would be useful to show the entire SCNA profile of the tumors in a heat map form, which is currently represented in Figure 2a.
- The investigators should explore whether cMET amplification occurs early upon dox withdrawal in KM tumors, which partially regress before they regrow. One would expect that cMET amplification occurs only in the re-growth phase of the tumor. This experiment might help determine whether a minor cMET⁺ clone is being selected for that should be resistant to apoptosis whereas neighboring cMET⁻ clones are actively regressing. Please see point below.
- While the authors present data that is highly suggestive that cMET amplification occurs de novo they still cannot convincingly demonstrate the preexistence of a minor clone with high levels of cMET that is positively selected upon dox withdrawal. Furthermore, cMET phosphorylation and genetic amplification might not necessarily be fully correlated. The gene might be amplified and primed for activation under selective pressure.
- The experiment in the Rag1^{-/-} brings up an interesting question that should be addressed. In these mice, cMET confers a growth advantage. If this is the case, why is cMET amplification not frequently seen in the initial primary tumors and thus selected for? In fact the initial set of experiments would suggest that cMET might not be beneficial until oncogene withdrawal after which it becomes beneficial. Could this discrepancy be due to the Rag1^{-/-} immune compromised context? If so the authors should transplant primary tumors prior to dox withdrawal into Rag1^{-/-} host to test the respective roles of Kras and cMET in the immune competent and immune compromised settings.
- Does the cMET inhibitor inhibit tumor growth in the presence of KRAS (aka. Without dox withdrawal)? This is an important control. Do any of the Kras⁺ tumors (aka without dox withdrawal) have cMET amplification? If so does the cMET inhibitor impact their growth?

Referee #2 (Remarks for Author):

Salgueiro and colleagues describe a phenomenon of chromosome instability (CIN) that they report as responsible for targeted therapy evasion. A mouse model of mammary cancer driven by KRAS with or without MAD2 overexpression, which causes CIN, is studied. In both models, a percentage of tumors avoids complete regression after KRAS expression is switched off. These non-regressing tumors had increased CIN, regardless of MAD2 status, but MAD2-overexpressing tumors relapsed faster and more often. Copy number analysis identified MET amplification in the majority of non-regressed tumors, resulting in increased MAD2 mRNA, protein, and protein phosphorylation. MET-amplified tumors responded to MET inhibitors *in vivo*. MET amplification may not have pre-existed, as untreated primary tumors did not appear to have any pMET-positive cells. These results suggest that a pre-existing CIN can accelerate resistance.

Overall, I find the study to be well-conducted and interesting. However, several major issues should be addressed.

Major points:

1. Fig. 1c: It would be helpful for authors to comment further on the differences in percentage of aberrations among tumor groups.
2. Fig. 2c: there should be a similar graph for IHC quantitation.
3. Phospho-cMet IHC was used as a means to assess MET amplification in 32 primary tumors, finding zero positive cells, but this is an indirect measurement that could be prone to false-negatives. Moreover, it is possible that MET amplification may not always lead to high pMet, especially in single cells that are potentially not in a high gradient of the Met ligand Hgf. FISH should be used as a more sensitive and direct test if authors want to truly rule out pre-existing MET amplification. It would be sufficient to test only the 3 primary tumor biopsies that later become MET-amp-positive non-regressed tumors.
4. The xenograft experiment is too short: even if 5 days of experiments are enough to prove some molecular effect, the actual therapeutic ability of the proposed treatment require at least 3 weeks of administration to be assessed. Moreover a clinically-proven drug such as crizotinib may be a better choice.

5. There should be at least a supplemental figure showing the full CN profiles of all chromosomes for all tumors (i.e. expanded version of Fig. 2b).

Minor points:

1. "Targeted therapy" has a specific connotation of small molecule or other pharmacological inhibitors. Since this study uses genetic KRAS extinction, it is suggested that the title be changed from "evade targeted therapy" to "circumvent oncogene addiction" or similar.
2. First paragraph of the results section should be re-structured as it is mainly relative to already published work. I'd suggest moving part of the paragraph to the introduction.
3. P6: "able to acquire a certain level of CIN to promote therapeutic resistance" - this phrasing presupposes that CIN drives resistance, when this hasn't yet been demonstrated at this point in the paper. I suggest replacing "to promote" with "associated with".
4. If possible, it would be useful to have Supplemental Table 1 become a main Table. It would also help to organize the tumors by their phenotype rather than by the ID number (i.e. group separately the MET+, KRAS re-expression, and "no obvious mechanism" sets).
5. Grammar should be revised by a native speaker.

Referee #3 (Comments on Novelty/Model System for Author):

The mouse model is adequate. Exploration of publicly available human tumor data is recommended, as detailed in my comments to the authors.

Referee #3 (Remarks for Author):

Salgueiro et al use a mouse model of Kras-driven mammary carcinoma to study chromosome instability (CIN) as a mechanism of therapeutic resistance, building up on their previous work (Rowald et al CellReports 2016). The main finding of the study is the high frequency of cMet amplification found in tumors that could grow independently of Kras - suggesting a mechanism of resistance to targeted therapy.

The findings are interesting and the conclusions are based on well-performed experiments with adequate controls. The paper is generally well written, although English / spelling requires some improvement throughout the text. The illustrations are of good quality and informative. Some adjustments in the general flow of the text may be needed to better fit the format of the Journal.

My specific comments to the authors are:

1. Introduction, Page 3. The last sentence "How prevalent..." should be refined as there are today several large cohorts of both early and metastatic breast cancer that have been genetically characterized, best examples come from Memorial Sloan Kettering in New York and from Institut Gustave Roussy in Paris. The most recent example is the paper by Bertucci et al Nature 2019.
2. Results, Page 5. The first paragraph of the results mainly refers to published data by the group (Rowald et al, 2016), as correctly cited in the text. This is a bit confusing, although I see the connection to the current manuscript. I suggest that the description of previously published data becomes shorter and/or is moved to the introduction.
3. Results, Page 6. It would be helpful if the authors described (perhaps in a diagram) the total number of tumors studied, so that the percentages and number of tumors with genetic alterations reported can be put into perspective.
4. Discussion. As suggested in comment 1 above, the authors are advised to look into the several datasets that are publicly available and see whether the observed genomic alterations (such as cMet amplifications) can also be seen in human tumors, either primary or metastatic (eg. In Bertucci et al Nature 2019 or in Zehir et al Nature Medicine 2017).
5. Discussion. The clinical implications of the findings should be further explored and discussed. What are the clinical data of cMET inhibitors? Is there room for improvement by better patient selection based on the current study? Has the high CIN proposed as a mechanism of resistance to targeted therapy seen in the clinic and if that is the case, should these tumors respond better to immunotherapy? (few data available but this does not seem to be the case)

Referee #1 (Remarks for Author):

In this manuscript, Salgueiro et al. use an elegant genetic model to study the effects of chromosomal instability on the ability of tumors to persist after oncogene withdrawal. Interestingly, they found that instability promoted tumor oncogene independence through the amplification of cMET. They then use a cMET inhibitor and found that non regressed tumors that amplified cMET were selectively sensitive to cMET inhibitors. Overall, this manuscript studies an important question and convincingly shows the importance of chromosomal instability in promoting resistance and explores a specific vulnerability that can be subsequently used to selectively kill adapted tumors. I support the publication of this manuscript after the authors have addressed the following points:

- It would be useful to show the entire SCNA profile of the tumors in a heat map form, which is currently represented in Figure 2a.

As requested by the reviewer we have added an extended view to show the full SCNA profile of the tumors, divided into the ones that contained the cMet amplification and the ones that did not. The data has been incorporated into Fig EV1B. The tumor numbers are described in Table 1.

- The investigators should explore whether cMET amplification occurs early upon dox withdrawal in KM tumors, which partially regress before they regrow. One would expect that cMET amplification occurs only in the re-growth phase of the tumor. This experiment might help determine whether a minor cMET+ clone is being selected for that should be resistant to apoptosis whereas neighboring cMET- clones are actively regressing. Please see point below.

We thank the reviewer for this suggestion. We have addressed this point by recapitulating the primary tumors in Rag2 immunocompromised animals, since it is faster than using the GEMM, which would need a minimum of 6-8 months. K and KM primary tumor cells were injected into the cleared fat pad. When tumors reached 1cm³, doxycycline food was changed to normal food to allow tumor regression. A few days later, when tumors were around 0.5cm³, they were collected and FISH and digital PCR (dPCR) was used to check if cMet was amplified. Both techniques gave similar results. We did not observe an increase in the amount of cMet copies. This data has been now incorporated into the manuscript (Fig 4).

- While the authors present data that is highly suggestive that cMET amplification occurs de novo they still cannot convincingly demonstrate the preexistence of a minor clone with high levels of cMET that is positively selected upon dox withdrawal. Furthermore, cMET phosphorylation and genetic amplification might not necessarily be fully correlated. The gene might be amplified and primed for activation under selective pressure.

As mentioned in the previous point, we performed FISH with a probe that recognizes mouse cMet in paraffin embedded tumor sections from 19 primary tumors. Although we found that K and KM primary tumors carried on average 6.22% and 5.67% respectively, of cells with more than 2 copies of cMet, the ratio of cMet probe/reference probe was close to 1 in all samples.

To verify these results with a different technique, we performed dPCR in 42 primary tumors using the TaqManTM CNV Assay for mouse cMet gene together with reference genes located in two different chromosomes TFRC: chromosome 16 or TERT: chromosome 13). The analysis was performed with the QuantStudio 3D AnalysisSuite Software. In concordance with the results from FISH, the ratio cMet: reference gene was close to 1 in 39 out of 42 tumors analyzed. In one of the tumors the ratio was 1.5 since it contained a whole chromosome 6 gain (PT KM15) and in case of PT K15 and KM4 the ratio was close to 1.5 using TFRC as reference, a result clearly explained by a whole chromosome 16 loss detected by WGS, where the reference gene TFRC is located.

This data has been incorporated to the manuscript (Fig 3E and Table 2).

- The experiment in the Rag1^{-/-} brings up an interesting question that should be addressed. In these mice, cMET confers a growth advantage. If this is the case, why is cMET amplification not frequently seen in the initial primary tumors and thus selected for?

We believe that cMet is not seen in the primary tumor since normally there is no need for two potent oncogenes to be expressed for form a tumor, and in fact in some cases this could even be detrimental

for the tumor cells. There is no need to activate the same signaling pathway mutating two oncogenes in the same pathway.

However, we have done an analysis of TCGA data looking for co-occurrence of MET amplification and KRAS mutation in Breast Cancer. This analysis shows that MET and KRAS activation (defining activation as either activation, overexpression over a defined threshold or activating point mutation) actually co-occur (p value= 0.021).

TCGA Breast Cancer

	KRAS ⁺	KRAS ⁻
MET ⁺	11	51
MET ⁻	76	822

p = 0.021
odds ratio = 2.33

In fact the initial set of experiments would suggest that cMET might not be beneficial until oncogene withdrawal after which it becomes beneficial. Could this discrepancy be due to the Rag1^{-/-} immune compromised context? If so the authors should transplant primary tumors prior to dox withdrawal into Rag1^{-/-} host to test the respective roles of Kras and cMET in the immune competent and immune compromised settings.

We addressed this question by comparing the time that tumor cells (either Kras primary cells or cMet amplified non-regressed cells) needed to form a tumor in immune deficient mice. We injected the same number of primary tumor cells from K or KM and non-regressed tumor cells (from K or KM with cMet amplification) and monitored the time these cells needed to form a tumor. The graph shows that cMet amplification induces significantly faster tumor development than Kras overexpression. Since this result is not the focus of the manuscript, whether non-regressed tumors with cMet grow faster than Kras primary tumors, we decided to not include this data into the manuscript, but show the results here for the reviewer.

- Does the cMET inhibitor inhibit tumor growth in the presence of KRAS (aka. Without dox withdrawal)? This is an important control. Do any of the Kras⁺ tumors (aka without dox withdrawal) have cMET amplification? If so does the cMET inhibitor impact their growth?

To evaluate the effect of the cMET inhibitor in Kras overexpressing tumors, we injected primary tumor cells into Rag2^{-/-} mice and set them on doxycycline. When tumors reached 0.5cm³, we treated the animals with Tepotinib or with vehicle control. We found no significant differences in tumor growth, between treated and untreated mice.

This data has been included in the manuscript (Fig 4).

Referee #2 (Remarks for Author):

Salgueiro and colleagues describe a phenomenon of chromosome instability (CIN) that they report as responsible for targeted therapy evasion. A mouse model of mammary cancer driven by KRAS with or without MAD2 overexpression, which causes CIN, is studied. In both models, a percentage of tumors avoids complete regression after KRAS expression is switched off. These non-regressing

tumors had increased CIN, regardless of MAD2 status, but MAD2-overexpressing tumors relapsed faster and more often. Copy number analysis identified MET amplification in the majority of non-regressed tumors, resulting in increased MAD2 mRNA, protein, and protein phosphorylation. MET-amplified tumors responded to MET inhibitors *in vivo*. MET amplification may not have pre-existed, as untreated primary tumors did not appear to have any pMET-positive cells. These results suggest that a pre-existing CIN can accelerate resistance. Overall, I find the study to be well-conducted and interesting. However, several major issues should be addressed.

Major points: 1. Fig. 1c: It would be helpful for authors to comment further on the differences in percentage of aberrations among tumor groups.

We have added additional information in the text.

2. Fig. 2c: there should be a similar graph for IHC quantitation.

We have quantified the percentages of cMet positive cells in the non-regressed tumors as requested by the reviewer. We have modified the figure to show all the non-regressed tumors analyzed. In addition, the percentages of positive cells are also shown in Table 1.

3. Phospho-cMet IHC was used as a means to assess MET amplification in 32 primary tumors, finding zero positive cells, but this is an indirect measurement that could be prone to false-negatives. Moreover, it is possible that MET amplification may not always lead to high pMet, especially in single cells that are potentially not in a high gradient of the Met ligand Hgf. FISH should be used as a more sensitive and direct test if authors want to truly rule out pre-existing MET amplification. It would be sufficient to test only the 3 primary tumor biopsies that later become MET-amp-positive non-regressed tumors.

This is a very good suggestion that was also requested by reviewer #1. To address this point, we performed FISH with a probe that recognizes mouse cMet in paraffin embedded tumor sections from 19 primary tumors (including the 3 biopsies that the reviewer is asking). Although we found that K and KM primary tumors carried on average 6.22% and 5.67% respectively, of cells with more than 2 copies of cMet, the ratio of cMet probe/reference probe was close to 1 in all samples. To verify these results with a different technique, we performed dPCR in 42 primary tumors using the TaqMan™ CNV Assay for mouse cMet gene together with the reference genes TFRC or TERT. The analysis was performed with the QuantStudio 3D AnalysisSuite Software. In concordance with the results from FISH, the ratio cMet: reference gene was close to 1 in 39 out of 42 tumors analyzed. In one of the tumors the ratio was 1.5 since it contained a whole chromosome 6 gain (PT KM15) and in case of PT K15 and KM4 the ratio is close to 1.5 since they contain a whole chromosome 16 loss, where the reference gene TFRC is located.

Additionally, we also performed FISH and dPCR in those K and KM primary tumors that were injected into Rag and harvested soon after doxycycline withdrawal. In both cases we found no detectable increase in the amount of cMet copies. This data has been now incorporated into the manuscript (Figure 3D-E, Figure 4 and Table 2).

4. The xenograft experiment is too short: even if 5 days of experiments are enough to prove some molecular effect, the actual therapeutic ability of the proposed treatment require at least 3 weeks of administration to be assessed. Moreover a clinically-proven drug such as crizotinib may be a better choice.

We agree with this reviewer and we have now performed the treatment with EMD1214063 (Tepotinib) for a longer period of time. However, we believe that the treatment with crizotinib is unnecessary since there are several studies showing the inefficacy of this drug against cMet positive tumors. In addition, crizotinib treatment could be challenging in these young immunocompromise mice since this drug is dissolved in an acidic vehicle and administered by oral gavage. We have experience with this drug treatment in immunocompetent mice in our lab and unfortunately many mice died because of the daily treatment. We were therefore more favorable of using EMD1214063 (also known as Tepotinib), which is currently in a phase II clinical trial (<https://clinicaltrials.gov/ct2/show/NCT02864992>).

We repeated the xenograft experiment to increase the treatment period. The non-regressed tumors without cMet amplification and with cMet amplification but treated with vehicle control behaved as previously, reaching human endpoint after 5-6 days. Interestingly, non-regressed tumors with cMet amplification that received Tepotinib underwent a period of regression that lasted for about 9 days. After this time, they resumed growth and after 19 days on drug treatment we had to sacrifice the mice. Altogether, this data suggests the development of a secondary mechanism of resistance. This data has been included in the manuscript (Figure 5).

5. There should be at least a supplemental figure showing the full CN profiles of all chromosomes for all tumors (i.e. expanded version of Fig. 2b).

As requested by the reviewer we have added an extended view to show the full SCNA profile of the tumors, divided into the ones that contained the cMet amplification and the ones that did not. The data has been incorporated into Fig EV1B. The tumor numbers are described in Table 1.

Minor points:

1. "Targeted therapy" has a specific connotation of small molecule or other pharmacological inhibitors. Since this study uses genetic KRAS extinction, it is suggested that the title be changed from "evade targeted therapy" to "circumvent oncogene addiction" or similar.

The title has been changed according to the reviewer's suggestion.

2. First paragraph of the results section should be re-structured as it is mainly relative to already published work. I'd suggest moving part of the paragraph to the introduction.

We have modified the results section and incorporated the already published work into the introduction.

3. P6: "able to acquire a certain level of CIN to promote therapeutic resistance" - this phrasing presupposes that CIN drives resistance, when this hasn't yet been demonstrated at this point in the paper. I suggest replacing "to promote" with "associated with".

This suggestion has been incorporated in P6.

4. If possible, it would be useful to have Supplemental Table 1 become a main Table. It would also help to organize the tumors by their phenotype rather than by the ID number (i.e. group separately the MET+, KRAS re-expression, and "no obvious mechanism" sets).

We have rearranged the table to show the data for cMet positive tumors and non cMet ones. The supplementary table is now Table 1. We also added another table with all the primary tumors used in the study, which is now table 2.

5. Grammar should be revised by a native speaker.

We apologized for the grammatical mistakes. The manuscript has now been revised by a native speaker.

Referee #3 (Comments on Novelty/Model System for Author):

The mouse model is adequate. Exploration of publicly available human tumor data is recommended, as detailed in my comments to the authors.

Referee #3 (Remarks for Author):

Salgueiro et al use a mouse model of Kras-driven mammary carcinoma to study chromosome instability (CIN) as a mechanism of therapeutic resistance, building up on their previous work (Rowald et al CellReports 2016). The main finding of the study is the high frequency of cMet

amplification found in tumors that could grow independently of Kras - suggesting a mechanism of resistance to targeted therapy.

The findings are interesting and the conclusions are based on well-performed experiments with adequate controls. The paper is generally well written, although English / spelling requires some improvement throughout the text. The illustrations are of good quality and informative. Some adjustments in the general flow of the text may be needed to better fit the format of the Journal.

My specific comments to the authors are:

1. Introduction, Page 3. The last sentence "How prevalent..." should be refined as there are today several large cohorts of both early and metastatic breast cancer that have been genetically characterized, best examples come from Memorial Sloan Kettering in New York and from Institut Gustave Roussy in Paris. The most recent example is the paper by Bertucci et al Nature 2019.

We have removed this sentence from the introduction since we now address this point in the discussion.

2. Results, Page 5. The first paragraph of the results mainly refers to published data by the group (Rowald et al, 2016), as correctly cited in the text. This is a bit confusing, although I see the connection to the current manuscript. I suggest that the description of previously published data becomes shorter and/or is moved to the introduction.

We have modified the results section and incorporated the already published work into the introduction.

3. Results, Page 6. It would be helpful if the authors described (perhaps in a diagram) the total number of tumors studied, so that the percentages and number of tumors with genetic alterations reported can be put into perspective.

The numbers have been included in the manuscript

4. Discussion. As suggested in comment 1 above, the authors are advised to look into the several datasets that are publicly available and see whether the observed genomic alterations (such as cMet amplifications) can also be seen in human tumors, either primary or metastatic (eg. In Bertucci et al Nature 2019 or in Zehir et al Nature Medicine 2017).

We added this point into the discussion.

5. Discussion. The clinical implications of the findings should be further explored and discussed. What are the clinical data of cMET inhibitors? Is there room for improvement by better patient selection based on the current study? Has the high CIN proposed as a mechanism of resistance to targeted therapy seen in the clinic and if that is the case, should these tumors respond better to immunotherapy? (few data available but this does not seem to be the case)

We added this point into the discussion.

2nd Editorial Decision

20 December 2019

Thank you for the submission of your revised manuscript to EMBO Molecular Medicine. We have now received the enclosed report from the referees who were asked to re-assess your work. As you will see, the referees are now supportive of publication. I am thus pleased to inform you that we will be able to accept your manuscript pending minor editorial amendments.

I look forward to reading a new revised version of your manuscript as soon as possible.

***** Reviewer's comments *****

Referee #1 (Remarks for Author):

The authors have adequately addressed my comments. I congratulate them on this revised version.

Referee #2 (Comments on Novelty/Model System for Author):

Great mouse model

Referee #2 (Remarks for Author):

Great work responding to the comments

Referee #3 (Remarks for Author):

The authors have addressed all comments and made significant improvements to the manuscript.

2nd Revision - authors' response

4 January 2020

Authors made the requested editorial changes.

Corresponding Author Name: Rocío Sotillo

Manuscript Number: EMM-2019-10941